# Interpretation of Multivariate Association Patterns between Multicollinear Physical Activity Accelerometry Data and Cardiometabolic Health in Children—A Tutorial

**DOI:** 10.3390/metabo9070129

**Published:** 2019-07-02

**Authors:** Eivind Aadland, Lars Bo Andersen, Geir Kåre Resaland, Olav Martin Kvalheim

**Affiliations:** 1Department of Sport, Food and Natural Sciences, Faculty of Education, Arts and Sports, Campus Sogndal, Western Norway University of Applied Sciences, 6856 Sogndal, Norway; 2Center for Physically Active Learning, Faculty of Education, Arts and Sports, Western Norway University of Applied Sciences, Campus Sogndal, 6856 Sogndal, Norway; 3Department of Chemistry, University of Bergen, 5007 Bergen, Norway

**Keywords:** multivariate pattern analysis, multiple linear regression, multicollinearity, statistics, cardiometabolic health, children, accelerometer, intensity

## Abstract

Associations between multicollinear accelerometry-derived physical activity (PA) data and cardiometabolic health in children needs to be analyzed using an approach that can handle collinearity among the explanatory variables. The aim of this paper is to provide readers a tutorial overview of interpretation of multivariate pattern analysis models using PA accelerometry data that reveals the associations to cardiometabolic health. A total of 841 children (age 10.2 ± 0.3 years) provided valid data on accelerometry (ActiGraph GT3X+) and six indices of cardiometabolic health that were used to create a composite score. We used a high-resolution PA description including 23 intensity variables covering the intensity spectrum (from 0–99 to ≥10000 counts per minute), and multivariate pattern analysis to analyze data. We report different statistical measures of the multivariate associations between PA and cardiometabolic health and use decentile groups of PA as a basis for discussing the meaning and impact of multicollinearity. We show that for high-resolution accelerometry data; considering all explanatory variables is crucial to obtain a correct interpretation of associations to cardiometabolic health; which is otherwise strongly confounded by multicollinearity in the dataset. Thus; multivariate pattern analysis challenges the traditional interpretation of findings from linear regression models assuming independent explanatory variables

## 1. Introduction

There is irrefutable evidence about the favorable influence of physical activity (PA) on cardiometabolic health [1]. However, this evidence is limited by sub-optimal handling of accelerometry data. Such data is commonly reduced to a spectrum of time spent in only few gross intensities (sedentary (SED), light PA (LPA), moderate PA (MPA), vigorous PA (VPA), and/or moderate-to-vigorous PA (MVPA)). Most studies investigating associations between PA and cardiometabolic health have targeted selected parts of this spectrum [1], partly because they rely on statistical methods that cannot handle multicollinear data. The focus on selected parts of the spectrum leads to a loss of resolution and thus information from accelerometry data. Hence, associations with metabolic health for the whole PA intensity spectrum needs to be addressed using appropriate methodology to obtain a better understanding of how PA relates to health [1,2,3,4].

Dependency and strong multicollinearity between intensity variables across the PA spectrum, represents a major limitation for common statistical methods such as ordinary least squares multiple linear regression [5]. Thus, statistical approaches that can overcome this challenge are needed [2,4,6]. The dependency among explanatory PA variables has been sought solved by the application of several alternative statistical approaches, including isotemporal substitution models [7,8], compositional data analysis [9,10], and multivariate pattern analysis [3,11,12]. Both the isotemporal substitution model and compositional data analysis take into account the closed structure of PA data, which means that since the total time budget is fixed all behaviors increase (or decrease) at the expense of others and time are therefore always reallocated among variables [4]. However, these approaches are special cases of multiple linear regression and do therefore not solve the multicollinearity challenge and do not allow for the analysis of more detailed descriptions of the PA intensity spectrum.

We have recently applied multivariate pattern analysis to examine associations for accelerometry data with cardiometabolic health in children [3,11,12]. Multivariate pattern analysis is widely applied in pharmaceutical [13] and metabolomics studies [14], in addition to other fields of biomedical research, such as in treatment and diagnosis of diseases [15], with the objective of revealing patterns of important biomarkers among hundreds or thousands of highly interrelated variables. As previously called for [1,2,6], this statistical method can handle completely collinear explanatory variables by combining the data into orthogonal latent variables [16]. Because multivariate pattern analysis solves the multicollinearity challenge, it allows for analyzing higher resolution accelerometry data (i.e., using a more detailed PA intensity profile than traditionally applied). Aadland et al. [3] applied 16 PA intensity intervals between 0–99 and ≥8000 counts per minute (cpm) (vertical axis), and found that intensities in the vigorous range (5000–7000 cpm) were strongest associated with cardiometabolic health. Moderate intensity PA was weakly related to health, while SED and LPA were not related to health. These findings can be further nuanced if using triaxial accelerometry [12], but we will not emphasize these findings and the added complexity herein.

The finding that VPA was more strongly associated with cardiometabolic health than other intensities [3], provide important information with regard to the development of PA guidelines. However, the interpretation of multivariate *association patterns* or *signatures* might not be straightforward. Specifically, the dose or duration of PA needed to achieve a certain health effect, which is obviously important for guideline development, is not evident from the paper by Aadland et al. [3]. The method used in the papers by Aadland et al. [3,11,12] was originally applied for biomarker discovery in chemical spectral profiles. Therefore, a “selectivity ratio” (SR), calculated as each explanatory variable’s ratio of explained to residual variance related to the predicted latent variable, was developed to reveal the strongest associations with the outcome among a high number of variables [17,18]. However, the SR is a statistical measure that is less known in the field of PA epidemiology, which challenges interpretation of these findings. Moreover, contrary to linear regression, which treats explanatory variables as independent to each other, multivariate pattern analysis determines how associations—in a multivariate space—relates to a given outcome. Since the explanatory accelerometry PA variables are strongly correlated [3], we regard this characteristic a strength of the method, but it also requires that interpretation of such results is based on patterns or reallocation of time across all the related explanatory variables that are important to the outcome. Since the derived variables have a closed structure within a finite period of time, behaviors substitute each other [4]. If sleep is included, the variables sum to 24 h; otherwise, variables sum to each individual’s accelerometer wear time (i.e., 100%). Thus, interpretation of a specific variable’s association to the outcome, includes changes across all explanatory variables. This covariation should be acknowledged as inherent in the data and a statistical model must be able to account for this feature.

The aim of the present paper is to provide readers a tutorial overview on interpretation of multivariate pattern analysis models of associations between accelerometry-derived PA and cardiometabolic health. We will specifically discuss how these models differ from commonly applied linear regression models and how association patterns from multivariate models can be used to inform guidelines for PA in relation to cardiometabolic health.

## 2. Materials and Methods 

### 2.1. Participants

The present study uses baseline data obtained from fifth-grade children in the Active Smarter Kids (ASK) cluster-randomized controlled trial, conducted in Norway during 2014–2015 [19,20]. Sixty schools, encompassing 1202 fifth-grade children, fulfilled the inclusion criteria, and agreed to participate. This sample represented 86.2% of the population of 10-year-olds in the county, and 95.2% of those eligible for recruitment. Later, three schools declined to participate. Thus, 1145 (97.4%) of 1175 available children from 57 schools agreed to participate in the study. 

Our procedures and methods conform to ethical guidelines defined by the World Medical Association’s Declaration of Helsinki and its subsequent revisions. The South-East Regional Committee for Medical Research Ethics in Norway approved the study protocol. We obtained written informed consent from each child’s parents or legal guardian and from the responsible school authorities prior to all testing. The study is registered in Clinicaltrials.gov with identification number: NCT02132494. 

### 2.2. Procedures

We have previously published a detailed description of the study [19], and therefore provide only a brief overview of the relevant procedures herein.

*Physical activity.* PA was measured using the ActiGraph GT3X+ accelerometer (Pensacola, FL, USA) [21]. Participants were instructed to wear the accelerometer at the waist at all times over seven consecutive days, except during water activities (swimming, showering) or while sleeping. Units were initialized at a sampling rate of 30 Hz. Files were analyzed at 1-second epochs to capture low and high intensity PA [11,22] using the KineSoft analytical software version 3.3.80 (KineSoft, Loughborough, UK). Data were restricted to hours 06:00 to 23:59. In all analyses, consecutive periods of ≥60 min of zero counts were defined as non-wear time [23]. We applied wear time requirements of ≥8 h/day and ≥4 days/week to constitute a valid measurement [24]. 

We created a dataset using 23 PA variables of total time (min/day) to capture movement in narrow intensity intervals throughout the intensity spectrum (vertical axis only); 0–99, 100–249, 250–499, 500–999, 1000–1499, 1500–1999, 2000–2499, 2500–2999, 3000–3499, 3500–3999, 4000–4499, 4500–4999, 5000–5499, 5500–5999, 6000–6499, 6500–6999, 7000–7499, 7500–7999, 8000–8499, 8500–8999, 9000–9499, 9500–9999, and ≥10000 cpm [12]. This approach is similar to the approach used by Aadland et al. [3], but extends the intensity spectrum in the vigorous intensity range. In addition, we used the Evenson et al. [25,26] cut points of 0–99, 100–2295, 2296–4011, ≥4012, and ≥2296 cpm to determine SED, LPA, MPA, VPA, and MVPA, respectively, and the proportion of children achieving the guideline PA level (mean of ≥60 min MVPA/day), for descriptive purposes. 

*Cardiometabolic health.* Aerobic fitness was measured with the Andersen intermittent running test, which has demonstrated acceptable reliability and validity in 10-year-old children [27]. Children ran as long as possible in a to-and-fro movement on a 20- meter track, touching the floor with a hand each time they turned, with 15-second work periods and 15-second breaks, for a total duration of 10 min. The distance (meters) covered was used as the outcome. Body mass was measured using an electronic scale (Seca 899, SECA GmbH, Hamburg, Germany) with children wearing light clothing. Height was measured using a portable Seca 217 (SECA GmbH, Hamburg, Germany). Body mass index (BMI) (kg·m^−2^) was calculated. Waist circumference was measured with a Seca 201 (SECA GmbH, Hamburg, Germany) ergonomic circumference measuring tape two cm over the level of the umbilicus. Systolic (SBP) and diastolic blood (DBP) pressures were measured using the Omron HBP-1300 automated blood pressure monitor (Omron Healthcare, Inc., Vernon Hills, IL, US). Children rested quietly for ten minutes in a sitting position with no distractions before blood pressures was measured four times; we used the mean of the last three measurements for analyses. Serum blood samples were collected from the children’s antecubital vein between 08:00 and 10:00 in the morning after an overnight fast. All blood samples were analyzed for total cholesterol (TC), triglyceride (TG), high-density lipoprotein cholesterol (HDL), glucose, and insulin at the accredited Endocrine Laboratory of the VU Medical Center (VUmc; Amsterdam, the Netherlands). Low-density lipoprotein cholesterol (LDL) was estimated using the Friedewald formula [28]. We calculated the TC:HDL ratio and homeostasis model assessment (HOMA) (glucose (mmol/L) * insulin (pmol/L) / 22.5) [29]. 

We calculated a composite score as the mean of six variables (SBP, TG, TC:HDL ratio, HOMA, waist:height ratio, and the reversed Andersen test) by averaging standardized scores after adjustment for sex and age. A higher score indicates increased risk, whereas a negative score indicates decreased risk. A similar approach has been used previously [3,30]. These composite score was used as the outcome in all models.

### 2.3. Statistical Analyses

Children’s characteristics were reported as frequencies, means, and standard deviations (SD). We tested for differences in characteristics between boys and girls using a linear mixed model to account for the clustering among studies. Models for PA and SED were adjusted for wear time. 

Partial least squares (PLS) regression analysis [16] followed by target projection [31] was used to determine the multivariate association pattern of PA with cardiometabolic health. We included all PA variables as explanatory variables and the composite cardiometabolic health score as the outcome variable, as shown previously [3]. PLS regression decomposes the explanatory variables into orthogonal linear combinations (PLS components), while simultaneously maximizing the covariance with the outcome variable. Thus, PLS regression is able to handle completely collinear variables [16]. Prior to PLS regression, all variables were centered and standardized to unit variance. Models were cross-validated using Monte Carlo resampling [32] with 1000 repetitions by repeatedly and randomly keeping 50% of the subjects as an external validation set when estimating the models. For each validated PLS regression model, a single predictive component was subsequently calculated by means of target projection [13,31] to express all the predictive variance in the PA intensity spectrum related to cardiometabolic health in a single intensity vector. SRs with 95% CIs were obtained as the ratio of this explained predictive variance to the residual variance for each PA intensity variable [17,18]. The procedure for obtaining the multivariate patterns is completely data-driven with no assumptions on variable distributions or degree of collinearity among variables. These analyses were performed by means of the commercial software Sirius version 11.0 (Pattern Recognition Systems AS, Bergen, Norway). 

*SR and alternative statistics.* The SR was developed as a statistic to separate important from less important information in mass spectral profiles acquired from cerebro-spinal fluid samples. The profiles were described by several thousands of mass-to-charge (m/z) intensities [17,18], measured as counts similar to accelerometry data. Thus, the development of SR was motivated by a need to highlight the signals with strongest associations to the outcome variable(s) after taking into account the noise level in the data. This measure, however, might be less informative for PA researchers and less directly interpretable compared to more common measures of association or effect size. The choice of statistic and its calculation and visualization is a matter of purpose. From the originally developed SR, given as the ratio of the explained to residual variance related to the target-projected component [17,18], several alternative statistics can be derived. In the present paper, we present four different alternatives: (1) Instead of dividing by the residual variance, we divided by the total variance, giving a measure of explained variance of each single explanatory variable, in a multivariate space, related to the target-projected component. This measure has a range of −1 to 1 when multiplying with the sign of the variable in the target-projected component. (2) Further, instead of relating this variance solely to the target projected component (i.e., the total explained variance related to the predicted outcome), we report a modified SR as the explained variance of the original outcome variable: SR * explained variance in the outcome. This statistic, thus, provides the explained variance of each of the explanatory PA variables for the actually measured outcome (instead of the predicted outcome). (3) Furthermore, by calculating the squared root of this statistic, we derived “multivariate correlation coefficients”, which are comparable to Pearson’s bivariate correlation coefficients (r), except that they are derived from the modelled part of the multivariate space. (4) As correlation coefficients are standardized to 1 SD, we finally weighted this correlation coefficient by each variable’s SD to obtain “unstandardized multivariate correlation coefficients” or coefficients of covariance. Thus, the statistics derived in step 3 and 4 are analogous to, but different from, standardized and unstandardized regression coefficients in linear regression. Using the latter (unstandardized) statistic allows for comparison of PA intensities’ importance for the outcome using actual units, that is, minutes per day. However, since the target projected component does not include so-called orthogonal variation among the explanatory variables and the outcome, these unstandardized multivariate correlation coefficients cannot be used directly for prediction of the outcome. For the purpose of prediction, the orthogonal variation needs to be accounted for in the model. Thus, the unstandardized multivariate correlation coefficients differ from regression coefficients.

*Interpretation of multivariate pattern models*. Since multivariate pattern analysis provides the association pattern—the signature—of how the explanatory variables in a multivariate space relate to the outcome, explanatory variables are not independent of each other. Thus, their separate association with the outcome cannot be determined from this analysis, which is reasonable given their strong interrelationships [3]. Furthermore, since time spent in each PA intensity replaces time spent in others, a meaningful interpretation must also incorporate reallocation of time as a basis for interpretation [4]. To illustrate this concept, we performed an analysis based on prediction of change in the composite cardiometabolic health score across sex-specific decentiles of PA using unstandardized multivariate correlation coefficients (covariances). As explained above, this measure provides the association with the composite score per 1 min/day change in a given intensity. For ease of interpretation, we merged PA intensities into the following 12 categories for this analysis: 0–99, 100–999, 1000–1999, 2000–2999, 3000–3999, 4000–4999, 5000–5999, 6000–6999, 7000–7999, 8000–8999, 9000–9999, and ≥10000 cpm. Boys and girls were separately categorized according to their age- and wear time-adjusted decentiles of time spent in the VPA intensity strongest related to cardiometabolic health (7000–7999 cpm) and a typical MPA (3000–3999 cpm) (40–43 children per group). These results are shown as the predicted changes in the cardiometabolic score according to each decentile group’s time spent in these intensity levels. To further simplify the understanding of patterns of change in PA across decentiles, we also summed time spent in 2000–3999 cpm and ≥4000 cpm for all decentiles, which roughly corresponds to the Evenson cut points [25] of MPA and VPA. Importantly though, since we show time use across intensities per 1000 cpm, any combination of intensities can be summed.

## 3. Results

### 3.1. Children’s Characteristics

We included 841 children (50% boys) who provided valid data on all relevant variables (Table 1). 

### 3.2. The Multivariate Association Pattern Displayed Using Different Statistics

In sum, the 23 PA variables explained 17.0% of the variance in cardiometabolic health. Figure 1 shows the association pattern between PA and cardiometabolic health using different statistics. All figures except the lower right figure, are based on standardized variables and are therefore virtually identical. The lower right figure is weighted by the variables’ SD and therefore lend more weight to higher intensities having a lower SD. As patterns were similar (r = 0.98), we merged boys and girls in these analyses.

### 3.3. Prediction of Cardiometabolic Health across Decentiles of PA

The unstandardized multivariate correlations for the PA intensity variables when merging the 23 variables to 12 variables are shown in Table 2. The coefficients are averaged using data from the lower right panel in Figure 1. The coefficients for 3000–3999 cpm (unstandardized r = −0.0453 min/day) and 7000–7999 cpm (unstandardized r = −0.4857 min/day) were used to predict the change in the cardiometabolic composite score across decentile groups for boys and girls separately. The pattern of time spent across the 12 intensities are shown in Table 3 and Table 4. Figure 2 and Appendix A show the predicted sex-specific cardiometabolic health scores for children across decentiles according to the difference between the time spent in 3000–3999 and 7000–7999 cpm and the group mean. For 3000–3999 cpm, it can be observed that the 10% least versus the 10% most active boys were predicted to have a difference in the composite score of 0.38 and −0.52 SDs compared to the average PA level, whereas the corresponding differences for girls were 0.44 and −0.26 SDs. For 7000–7999 cpm, it can be observed that the 10% least versus the 10% most active boys were predicted to have a difference in the composite score of 1.08 and −1.48 SDs compared to the average PA level, whereas the corresponding differences for girls were 0.79 and −0.99 SDs.

## 4. Discussion

In the present study, we address challenges with regard to the interpretation of associations between multicollinear accelerometry-derived PA variables and cardiometabolic health in children. Contrary to findings from commonly-applied linear regression models, where associations are interpreted as independent of each other, associations from multivariate pattern analysis are derived from the joint pattern of all explanatory variables and must therefore be interpreted differently. In the following sections, we will discuss these differences and attempt to provide readers a framework of how findings from multivariate pattern analysis can be applied to inform guidelines for PA.

Many studies in the field of PA epidemiology include only few of the many potentially important accelerometry-derived PA variables. This practice substantially reduces the information about the influence of various intensities on cardiometabolic health and it increase susceptibility to residual confounding [1,2,3,12]. For example, most studies merge all intensities above MPA as MVPA, which gives the same weight to brisk walking and fast running. Thus, we and others [1,2,3,4,12] argue that associations with cardiometabolic health for the whole PA intensity spectrum should be addressed to obtain a complete picture and facilitate a better understanding of how PA relates to cardiometabolic health. Indeed, we have shown that the explained variance improves up to tenfold when adding higher resolution data compared to traditionally applied overall summary measures (explained variance = 3.2, 4.8, 17.0, and 30.3% for overall cpm from the vertical axis, MVPA from the vertical axis, the whole intensity spectrum from the vertical axis, and the whole intensity spectrum from triaxial accelerometry, respectively) [12]. However, due to the strong multicollinearity between variables, which multiple linear regression cannot handle [5], we need statistical methods that overcome this challenge [4,6]. Aadland et al. [3,11,12] have previously addressed the collinearity challenge of accelerometry-derived PA data using multivariate pattern analysis, which can treat accelerometry-derived PA variables as an intensity spectrum without limitations regarding the number and distributions of variables being analyzed and without any transformation of data [13,15,16]. However, it complicates the interpretation of associations, since the information about the outcome is derived from many explanatory variables that must be treated jointly.

Our findings suggest all relevant information for the outcome can be accounted for by PA intensities in the moderate to vigorous area. In such a case, including one variable—MVPA—as the explanatory variable, means that its coefficient can be used directly to state that a change of 1 min/day is associated with for example −0.3059 SDs change in the cardiometabolic composite score (i.e., the mean of all unstandardized coefficients ≥2000 cpm from Table 2). On the contrary, a coefficient of −0.4857 for 7000–7999 cpm differs substantially in terms of its interpretation. Spending 1 min/day more in 7000–7999 cpm does not improve cardiometabolic health by approximately 0.5 SDs per se. As seen from Table 4, differences across decentiles in one variable relates strongly to differences across decentiles in other proximal variables. Thus, the difference in time spent in 7000–7999 cpm between for example decentile 3 and 8 of 2.13 min/day for boys, is inseparable from the differences of 4.1, 3.5, 3.0, 1.3, 0.9, and 3.9 min/day for 4000–4999, 5000–5999, 6000–6999, 8000–8999, 9000–9999, and ≥10000 cpm. In sum, the difference in VPA (≥4000 cpm) between these decentile groups amounts to 18.9 min/day (32.2 vs. 51.1 min/day). The coefficient for 7000–7999 cpm incorporates this pattern and is, in comparison with the other (standardized) variable estimates, the variable strongest associated with cardiometabolic health. An even more extreme example is the association of −0.8522 for 9000–9999 cpm, which means spending 1 min/day more in this intensity relates to almost 1 SD improvement in cardiometabolic health. However, the difference of 1.00 min/day for girls in decentile 2 versus 9 (Table 4) is inherently part of a 13.9 min/day (20.8 vs. 34.7 min/day) difference in VPA in total (≥4000 cpm). In contrast to using for example MVPA as a single variable that grossly captures all relevant information with regard to the outcome, associations for variables from a higher resolution dataset are clearly not interpretable separately because such models are strongly confounded by other proximal variables. The association of −0.4857 for 7000–7999 cpm as analyzed using a simple linear regression model would clearly suffer from severe residual confounding, which cannot be corrected, because the confounding variables are intrinsically part of the same pattern and thus multicollinear to the focused explanatory variable. This point is also illustrated by the non-addable nature of the coefficients; while the model in total explained 17.0% of the variation in the outcome, each variable between 6000 and 8999 cpm seemingly explained ≥10% of the outcome (Figure 1, upper right panel) and all variables in sum explained >100% of the outcome. Obviously, these contrasting interpretations is caused by the strong multicollinearity and thus great redundancy of information among variables.

Although the signature of PA associated with cardiometabolic health primarily is characterized by VPA, which clearly informs PA guideline development with regard to the importance of higher PA intensities for improved health, the question regarding how much time children optimally should spend in diverse PA intensities is not easily answered. By deriving unstandardized coefficients, we can directly compare different intensity variables’ strengths of association to health. We previously reported that time spent in VPA (5000–7999 cpm) was 5 times more important for cardiometabolic health than MPA (2500–3500 cpm) using 16 PA spectrum variables and 10-second epoch data [3]. In the present study, using 23 PA spectrum variables and 1-second epoch data, we show that unstandardized associations (per 1 min/day) for 7000–7999 cpm (coefficient −0.4857) are 10 times stronger than for 3000–3999 cpm (coefficient −0.0453) (means for any summation of data can be calculated from Table 2). However, as these associations, as discussed in the previous paragraph, must be interpreted with the whole spectrum of intensities as the backdrop, it is still impossible to obtain a clear understanding about how duration of single PA intensities relates to health (or other outcomes). In an attempt to clarify this aspect, we categorized and compared children across decentiles of MPA (3000–3999 cpm) and the VPA intensity strongest associated with cardiometabolic health (7000–7999 cpm). Worth noting, contrary to what should be expected from the difference in coefficients, the influence on cardiometabolic health of change in MPA and VPA (on average −0.08 and −0.21 SDs per decentile, respectively) differ only 2–3 times in favor of VPA. However, this smaller difference is caused by incomparable differences in time spent across decentiles for MPA and VPA; for example, while the difference in time spent in the most extreme decentiles for boys is 19.8 min/day for 3000–3999 cpm, it is only 5.26 min/day for 7000–7999 cpm. Nevertheless, because children (or adults) do not exercise in narrow intensity intervals, further simplification using gross intensity zones is necessary to inform and message guidelines. Based on the associations given in Table 2, it is evident that a 1 min/day higher PA level in 7000–7999 cpm is associated with an approximately 0.5 SD lower cardiometabolic health composite score. Spending 1 min/day more in 7000–7999 amounts to approximately 10 min/day more in VPA in total (9.1 and 10.0 min/day in boys and girls, respectively) and approximately 15 min/day in MVPA in total (13.0 and 15.5 min/day in boys and girls, respectively). Thus, our results suggest that increasing VPA by 10 min/day, or if further simplification is warranted; MVPA by 15 min/day, relates to an approximately 0.5 SD improved cardiometabolic health score in 10 year old children.

### Strengths and Limitations

The main strength of the present study is the use of a dataset with a large sample of children, which allowed for determination of robust and stable association patterns and comparisons across decentile groups. Furthermore, an important strength of our approach using the whole PA intensity spectrum is that application of pre-defined accelerometer intensity cut points is not necessary. Because cut points vary considerably between studies [33], they hamper the interpretation of results regarding the different PA intensities’ importance for health. If, for example, we consider two influential studies in the field; Andersen et al. [30] defined MVPA above 2000 cpm and Ekelund et al. [34] defined MVPA above 3000 cpm. Such variation might easily confuse findings and comparability among studies. Thus, using the whole intensity spectrum provide a more nuanced and robust picture of the associations between PA and cardiometabolic health.

Because our analyses were restricted to cross-sectional associations, a limitation of our study is that we could not infer causality from our findings. Moreover, as only one dataset of relatively active Norwegian children and one outcome (although a composite score of cardiometabolic health) were used for the analysis, future studies are needed to extend our work using a similar analytic approach applied to other datasets including various samples and outcomes. Although we have shown previously that the multivariate PA intensity signature of PA is rather similar for adiposity, cholesterol, lipids, insulin sensitivity, and aerobic fitness [3], evidence of associations with other outcomes, for which the association pattern might be different, are warranted. Moreover, association patterns corresponding to those shown herein should also be established using accelerometry data obtained from other placements, for example the wrist, and for raw acceleration data, which circumvent the challenge of brand-specific interpretation of accelerometer “counts”. Because the placement of the accelerometer will affect which activities that are captured at a certain acceleration or count level, association signatures might differ across placements, consistent with our findings for triaxial accelerometry, where different axes captures different characteristics of PA [12].

Finally, although objective monitoring of PA by means of accelerometry has lead to significant progress in the field, such measurements have some limitations. Many activities, for example swimming, upper body movements, and cycling, are poorly captured by the accelerometer [35,36]. Moreover, there are great individual variation in the intensity level at a given count level, especially at high intensity/count levels. For example, Evenson et al. [25] found SDs of 2328 and 4280 cpm for running at 6.5 km/h (mean 4700 cpm) and jumping jacks (mean 9496 cpm), respectively. The physical effort of children performing these activities will obviously also differ largely depending on their physical fitness level. Nevertheless, the PA intensities most strongly associated with health (7000–7999 cpm) herein, which based on Evenson et al.’s [25] findings is placed between running and jumping jacks, would clearly be vigorous intensities (oxygen consumption 26.5–28.1 mL/kg/min; 7.6–8.0 metabolic equivalents [25]), which is also supported by Trost et al. [26], concluding that cut points of approximately 4000 cpm provide the best classification accuracy of VPA. Thus, consistent with our simplified public health message above, our results suggest children should perform activities that involve running and jumping to improve cardiometabolic health.

## 5. Conclusions

Multivariate pattern analysis has the ability to model simultaneously multiple highly correlated variables. As applied to accelerometry data, it uses and treats all available information together, resulting in stronger and stable models of *patterns* of associations between PA variables and cardiometabolic outcomes [3,11,12]. In this paper, we aimed to provide a tutorial overview and discussion of how such association patterns can be reported and interpreted, with specific emphasis on the meaning of multicollinearity of higher resolution accelerometry data. Although a certain level of prior statistical knowledge may help in understanding this approach, we have focused our discussion on the conceptualization of such data as a pattern of inseparable explanatory variables. Still, a meaningful translation of findings using such methodology is necessary to inform PA guidelines, because guidelines are not messaged based on narrow intensity intervals and high-resolution data. We have suggested one approach to meet this goal, but further research is warranted to extend our findings and perspectives.

## Figures and Tables

**Figure 1 metabolites-09-00129-f001:**
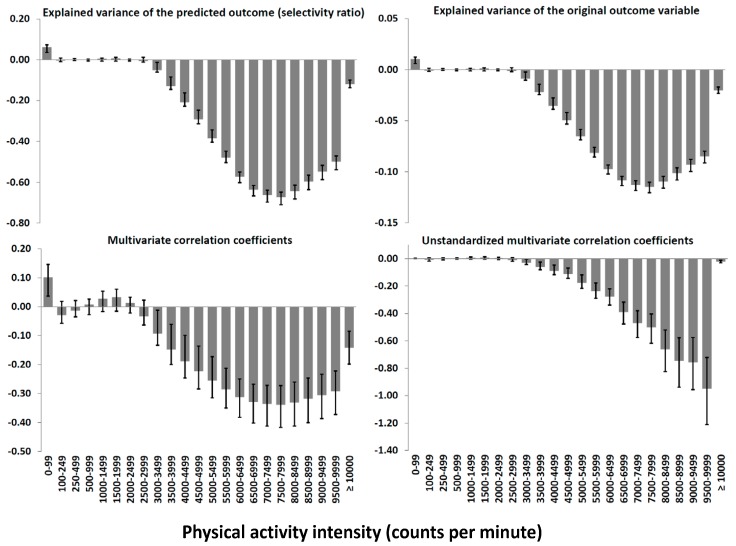
The multivariate association pattern between physical activity and cardiometabolic health reported using different statistics.

**Figure 2 metabolites-09-00129-f002:**
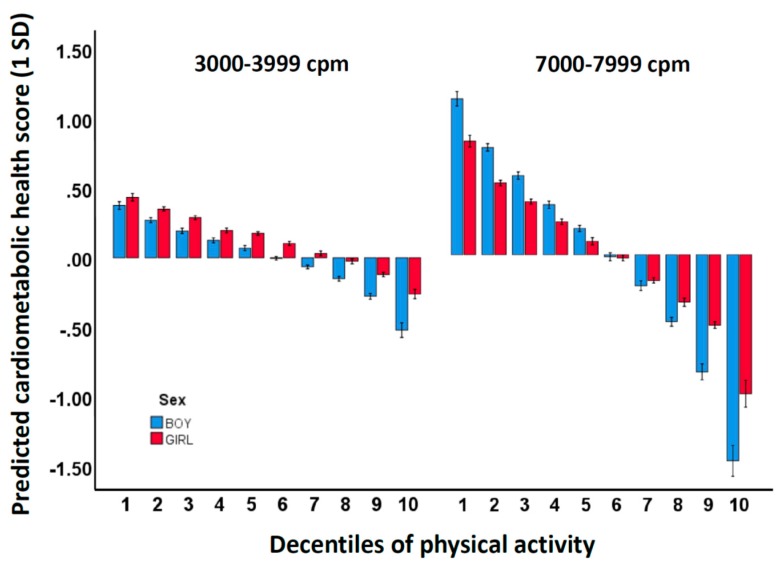
The predicted change in cardiometabolic health as compared to the average physical activity level in the various decentile groups.

**Table 1 metabolites-09-00129-t001:** Children’s characteristics.

	Overall (n = 841)	Boys (n = 424)	Girls (n = 417)	*p* between Groups
**Demography**				
Age (years)	10.2 (0.3)	10.2 (0.3)	10.2 (0.3)	0.803
**Anthropometry**				
Body mass (kg)	37.0 (8.1)	36.8 (7.8)	37.2 (8.3)	0.641
Height (cm)	142.9 (6.7)	143.1 (6.7)	142.6 (6.8)	0.197
BMI (kg/m^2^)	18.0 (3.0)	17.9 (2.9)	18.1 (3.1)	0.218
Overweight and obese (%)	20.8	20.0	21.5	0.583
Waist circumference (cm)	61.9 (7.5)	62.2 (7.3)	61.6 (7.7)	0.169
Waist:height (ratio)	0.43 (0.05)	0.43 (0.05)	0.43 (0.05)	0.322
**Indices of cardiometabolic health**				
Andersen test (m)	898 (103)	925 (112)	871 (85)	<0.001
Systolic blood pressure (mmHg)	105.2 (8.4)	105.3 (8.2)	105.2 (8.6)	0.612
Diastolic blood pressure (mmHg)	57.7 (6.2)	57.4 (6.0)	58.1 (6.3)	0.180
Total cholesterol (mmol/l)	4.46 (0.69)	4.46 (0.70)	4.46 (0.68)	0.976
LDL-cholesterol (mmol/l)	2.51 (0.64)	2.50 (0.65)	2.53 (0.62)	0.570
HDL-cholesterol (mmol/l)	1.59 (0.35)	1.63 (0.34)	1.55 (0.35)	0.001
Total:HDL-cholesterol (ratio)	2.91 (0.71)	2.82 (0.66)	2.99 (0.74)	0.001
Triglyceride (mmol/l)	0.78 (0.38)	0.72 (0.31)	0.84 (0.42)	<0.001
Glucose (mmol/l)	4.98 (0.32)	5.02 (0.31)	4.94 (0.33)	0.001
Insulin (pmol/l)	7.91 (4.29)	7.05 (3.48)	8.33 (4.83)	<0.001
HOMA (index)	1.71 (0.98)	1.54 (0.83)	1.89 (1.09)	<0.001
Composite score (1SD) *	0.00 (1.00)	0.00 (0.93)	0.00 (1.07)	-
**Physical activity**				
Wear time (min/day)	795 (56)	799 (59)	791 (54)	0.032
Overall physical activity (cpm)	708 (272)	754 (296)	660 (235)	<0.001
SED (min/day)	597 (56)	593 (59)	601 (53)	<0.001
LPA (min/day)	122 (22)	124 (23)	120 (21)	0.065
MPA (min/day)	37 (10)	39 (10)	35 (8)	<0.001
VPA (min/day)	39 (15)	43 (16)	35 (12)	<0.001
MVPA (min/day)	76 (23)	82 (24)	70 (19)	<0.001
Guideline amount (%)	74	80	68	<0.001

BMI = body mass index; LDL = low density lipoprotein; HDL = high density lipoprotein; HOMA = homeostasis model assessment; SED = sedentary time; LPA = light physical activity; MPA = moderate physical activity; VPA 0 vigorous physical activity; MVPA = moderate-to-vigorous physical activity. * The composite score includes waist:height ratio, systolic blood pressure, total:HDL ratio, triglycerides, HOMA, and the Andersen test. Intensity-specific PA is calculated using the Evenson cut points [25]; The guideline PA levels is defined as a mean of ≥60 min of MVPA per day.

**Table 2 metabolites-09-00129-t002:** Regression coefficients (unstandardized) obtained from the multivariate pattern analysis.

Physical Activity Intensity (cpm)	Unstandardized Multivariate Regression Coefficients
0–99	0.0018
100–999	−0.0038
1000–1999	0.0068
2000–2999	−0.0034
3000–3999	−0.0453
4000–4999	−0.1005
5000–5999	−0.2070
6000–6999	−0.3337
7000–7999	−0.4857
8000–8999	−0.7040
9000–9999	−0.8522
≥10000	−0.0217

Coefficients represent a simplification of the pattern shown in the lower right panel of Figure 1 calculated by averaging coefficients across intervals to reduce the number of PA variables from 23 to 12.

**Table 3 metabolites-09-00129-t003:** Mean time (min/day) spent in PA intensities according to decentiles of time spent in 3000–3999 cpm.

Physical Activity Intensity (cpm)	Decentiles of 3000–3999 cpm										
	1	2	3	4	5	6	7	8	9	10	Mean
**Boys**											
0–99	660	628	624	604	583	590	564	566	566	545	593
100–999	60.5	65.2	65.8	69.0	69.7	73.4	71.4	71.9	77.2	77.8	70.2
1000–1999	34.0	38.8	39.5	42.5	43.1	44.1	46.1	46.7	50.9	52.7	43.9
2000–2999	19.6	22.7	24.1	26.3	27.6	28.3	30.3	32.2	35.4	38.6	28.5
3000–3999	**12.2**	**14.5**	**16.2**	**17.7**	**19.0**	**20.6**	**21.9**	**23.8**	**26.6**	**32.0**	20.5
4000–4999	7.3	9.1	10.3	11.4	12.1	13.4	14.2	15.6	16.7	21.2	13.1
5000–5999	4.6	5.8	6.5	7.4	7.7	8.6	8.9	9.7	10.1	12.4	8.2
6000–6999	3.4	4.2	4.8	5.5	5.5	6.2	6.3	6.9	7.2	8.2	5.8
7000–7999	2.28	2.74	3.14	3.66	4.06	4.06	4.14	4.36	4.62	5.04	3.77
8000–8999	1.45	1.69	1.91	2.31	2.27	2.46	2.61	2.65	2.83	2.91	2.31
9000–9999	1.05	1.19	1.37	1.67	1.61	1.73	1.91	1.86	1.99	2.03	1.64
≥10000	4.18	5.30	6.34	7.15	8.43	8.82	10.81	11.03	9.22	10.06	8.14
2000–3999	31.8	37.2	40.3	44.0	46.6	48.9	52.2	56.0	62.0	70.6	49.0
≥4000	24.3	30.0	34.4	39.1	41.7	45.3	48.9	52.1	52.7	61.8	43.0
**Girls**											
0–99	653	630	615	616	598	585	600	576	584	555	601
100–999	62.2	65.0	65.8	72.8	65.1	67.7	71.9	73.9	74.0	75.7	69.4
1000–1999	33.6	35.6	38.6	42.0	39.7	42.1	43.3	46.0	46.1	48.3	41.5
2000–2999	17.9	20.2	22.4	24.6	24.8	27.2	27.9	29.6	31.2	34.1	26.0
3000–3999	**10.9**	**12.8**	**14.1**	**16.2**	**16.6**	**18.2**	**19.8**	**21.6**	**23.2**	**26.3**	17.9
4000–4999	6.5	7.9	8.5	9.8	10.2	11.0	11.9	13.0	14.1	16.2	10.9
5000–5999	3.9	5.0	5.3	6.0	6.2	6.7	7.0	7.7	8.1	9.3	6.5
6000–6999	2.8	3.7	3.8	4.4	4.4	4.7	5.0	5.4	5.5	6.3	4.6
7000–7999	1.81	2.41	2.59	2.99	2.87	3.05	3.29	3.47	3.61	3.94	3.00
8000–8999	1.12	1.51	1.61	1.90	1.78	1.87	2.03	2.13	2.25	2.43	1.86
9000–9999	0.82	1.09	1.17	1.35	1.28	1.34	1.48	1.54	1.61	1.72	1.34
≥10000	3.79	6.16	5.39	6.95	7.20	6.79	8.22	9.13	9.07	8.80	7.15
2000–3999	28.8	33.0	36.5	40.8	41.4	45.4	47.7	51.2	54.4	60.4	43.9
≥4000	20.7	27.8	28.4	33.4	33.9	35.5	38.9	42.4	44.2	48.7	35.4

The variable in bold indicates the basis for construction of decentiles.

**Table 4 metabolites-09-00129-t004:** Mean time (min/day) spent in PA intensities according to decentiles of time spent in 7000–7999 cpm.

Physical Activity Intensity (cpm)	Decentiles of 7000–7999 cpm										
	1	2	3	4	5	6	7	8	9	10	Mean
**Boys**											
0–99	647	624	605	602	586	586	573	583	569	554	593
100–999	61.1	68.2	71.6	69.7	72.2	68.6	70.0	73.4	73.5	73.5	70.2
1000–1999	36.5	41.5	42.9	43.6	45.3	42.7	44.6	46.2	47.3	48.1	43.9
2000–2999	22.9	26.0	27.5	27.6	29.1	27.8	30.0	30.0	31.3	33.0	28.5
3000–3999	15.1	17.2	18.5	19.5	20.9	20.1	22.1	22.2	23.5	25.8	20.5
4000–4999	8.2	10.0	10.8	12.4	13.3	12.9	14.4	14.9	16.1	18.4	13.1
5000–5999	4.4	5.7	6.2	7.3	8.0	8.0	9.1	9.7	10.7	12.5	8.2
6000–6999	2.8	3.7	4.2	4.8	5.4	5.8	6.5	7.2	8.2	9.8	5.8
7000–7999	**1.55**	**2.23**	**2.59**	**3.02**	**3.41**	**3.77**	**4.20**	**4.72**	**5.45**	**6.81**	3.77
8000–8999	0.91	1.33	1.55	1.83	2.07	2.32	2.57	2.87	3.36	4.28	2.31
9000–9999	0.64	0.94	1.09	1.31	1.50	1.66	1.79	2.02	2.38	3.07	1.64
≥10000	2.83	4.21	5.80	6.57	8.33	8.59	8.29	9.72	12.16	14.97	8.14
2000–3999	38.0	43.2	46.0	47.1	50.0	47.9	52.1	52.2	54.8	58.8	49.0
≥4000	21.3	28.1	32.2	37.2	42.0	43.0	46.9	51.1	58.4	69.8	43.0
**Girls**											
0–99	646	624	620	607	597	587	604	590	567	567	601
100–999	61.6	66.7	69.8	66.3	71.0	66.4	71.4	72.2	72.1	76.1	69.4
1000–1999	34.9	38.9	40.4	38.6	42.6	40.8	42.2	43.8	48.2	20.7	41.5
2000–2999	20.7	23.2	24.4	23.7	26.6	26.4	27.0	28.7	27.4	31.8	26.0
3000–3999	13.2	15.2	16.4	16.5	18.0	18.2	19.2	19.9	19.5	23.1	17.9
4000–4999	7.2	8.7	9.6	10.0	10.7	11.1	11.8	12.3	12.6	15.1	10.9
5000–5999	3.8	4.9	5.6	5.8	6.5	6.8	7.1	7.6	7.8	9.7	6.5
6000–6999	2.4	3.2	3.7	3.9	4.5	4.7	5.1	5.5	5.8	7.2	4.6
7000–7999	**1.38**	**1.93**	**2.27**	**2.48**	**2.82**	**3.05**	**3.34**	**3.70**	**4.01**	**5.05**	3.00
8000–8999	0.82	1.20	1.33	1.50	1.72	1.89	2.05	2.32	2.58	3.24	1.86
9000–9999	0.58	0.87	0.96	1.07	1.22	1.36	1.45	1.71	1.87	2.30	1.34
≥10000	2.50	5.01	5.06	4.84	7.16	6.47	7.15	9.49	11.50	12.22	7.15
2000–3999	33.9	38.4	40.8	40.2	44.6	44.6	46.2	48.6	46.9	54.9	43.9
≥4000	16.2	20.8	23.5	24.8	27.5	28.9	30.8	33.1	34.7	42.6	28.2

The variable in bold indicates the basis for construction of decentiles.

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
