# Peer review of "Interpretation of Multivariate Association Patterns between Multicollinear Physical Activity Accelerometry Data and Cardiometabolic Health in Children—A Tutorial"

_metabolites, 2019, doi:10.3390/metabo9070129_

Reviewer 1 Report

Thanks for the opportunity to review this manuscript which provides a tutorial style breakdown of the multivariate pattern analysis method that has recently been applied by the authors to accelerometer data. I have to say that I’m not a statistics expert and so cannot comment on the technical elements of the analysis approach with any great authority. Therefore, I have approached the review from an applied standpoint and my comments hopefully will reflect this, and help to make the methods described as clear as possible for researchers who may consider using them.

Lines 39-40. Compositional data analysis is another analytical approach that has become more prominent in recent years, and which also accounts for multi-collinearity in accelerometer PA data. This approach though also relies on the PA data being expressed as ‘gross intensities’. Reference to this the compositional data analysis approach would be a useful addition at this point in the manuscript to illustrate how a range of new/different approaches are now being used.

Lines 116-118. I get that the Evenson cutpoints were used for illustrative purposes, and this is helpful. However, perhaps it might also be appropriate to highlight that with the multivariate pattern analysis method, cutpoints are not needed as the full intensity range of the counts data is being used.

Table 1. The children in the sample were very active based on the Evenson cutpoints (74% meeting guideline). I’m wondering whether the strength of associations and range of intensities would be the same for a low active sample, and whether any sub-group analyses were done to look at this.

General comments

Have the authors applied the multivariate pattern analysis approach with other health outcomes, such as muscle/bone health, psychosocial wellbeing, etc? Can they comment on the applicability of the approach with other outcomes other than cardiometabolic health?

Please can the authors comment on the applicability of the multivariate pattern analysis approach for accelerometer data derived from wear sites other than the hip. Would the pattern of associations be expected to change?

The studies using the multivariate pattern analysis approach published to date plus this tutorial have used ActiGraph counts data. As more studies use data from other accelerometer brands which is often based on raw accelerations rather than counts, can the authors comment on the applicability of the approach in these situations?         

Syntax

Line 36. “measurements”

Line 204. “provides”

Line 240. “coefficients”

Line 241. “were used”

Line 244. “shows”

Line 249. “boys were predicted to”

Line 273. Is it “some” or “few”?

Line 280. “improves”

Line 365. “guidelines are not”

Reviewer 2 Report

I read manuscript "Interpretation of multivariate association patterns between multicollinear physical activity accelerometry data and cardiometabolic health in children – a tutorial" by Dr. Aadland and co-workers. The paper describes methodology how to improve accelerometry data analysis and interpretation and its relationship to cardiometabolic risk. The paper was well written and provided a nice overview flaws and new opportunities for accelerometry data analysis. I have only  few comments and questions for the Authors.

1. The multivariate pattern analysis provides several new approaches to better understand the associations of physical activity with cardiometabolic risk in youth. Altough this approach has several strenghts, it does not remove the problems related to fixed intensity cut offs. i.e. altough it is possible that the authors find that  7000cpm is the strongest correlate of cardiometabolic risk, it is still not possible to know how intense such physical activity is for an individual. Some youth may not ever achieve such values while for some it may be moderate because their level of fitness is much higher. I suggest that the Authors comment this issue.

2. The Authors include Andersen test result as a measure of cardiorespiratory fitness as one component of cardiometabolic risk. Using it as a "powerful marker of health" has of course historical burden, but it is possible than field based tests of fitness as well as VO2max normalised for body mass overemphasise importance of fitness in health outcomes in youth (e.g. doi:10.1001/jamapediatrics.2019.1485). Another question is that if you include a measure of fitness in cardiometabolic risk score, how it will affect on the associations between accelerometry metrics and cardiometabolic health. It is possible that high fit children accumulate higher count values than lower fit children because it is easier for them (but in fact there may not be difference in physiological intensity, such as VO as % of VO2 reserve, between those children). This is worth of discussing.
